# Membrane Proteomic Profiling of Soybean Leaf and Root Tissues Uncovers Salt-Stress-Responsive Membrane Proteins

**DOI:** 10.3390/ijms232113270

**Published:** 2022-10-31

**Authors:** Hafiz Mamoon Rehman, Shengjie Chen, Shoudong Zhang, Memoona Khalid, Muhammad Uzair, Phillip A. Wilmarth, Shakeel Ahmad, Hon-Ming Lam

**Affiliations:** 1Centre for Soybean Research of the State Key Laboratory of Agrobiotechnology and School of Life Sciences, The Chinese University of Hong Kong, Hong Kong, China; 2Centre of Agricultural Biochemistry and Biotechnology (CABB), University of Agriculture Faisalabad, Faisalabad 38000, Pakistan; 3Department of Biochemistry & Cellular and Molecular Biology, University of Tennessee, Knoxville, TN 37996, USA; 4Proteomics Shared Resource, Oregon Health & Science University, 3181 SW Sam Jackson Park Road, Portland, OR 97239, USA; 5Seed Center, Ministry of Environment, Water & Agriculture, Riyadh 14712, Saudi Arabia

**Keywords:** salt stress, membrane proteome, soybean, orbitrap, stress-inducible proteins, label-free quantification, mass spectrometry

## Abstract

Cultivated soybean (*Glycine max* (L.)), the world’s most important legume crop, has high-to-moderate salt sensitivity. Being the frontier for sensing and controlling solute transport, membrane proteins could be involved in cell signaling, osmoregulation, and stress-sensing mechanisms, but their roles in abiotic stresses are still largely unknown. By analyzing salt-induced membrane proteomic changes in the roots and leaves of salt-sensitive soybean cultivar (C08) seedlings germinated under NaCl, we detected 972 membrane proteins, with those present in both leaves and roots annotated as receptor kinases, calcium-sensing proteins, abscisic acid receptors, cation and anion channel proteins, proton pumps, amide and peptide transporters, and vesicle transport-related proteins etc. Endocytosis, linoleic acid metabolism, and fatty acid biosynthesis pathway-related proteins were enriched in roots whereas phagosome, spliceosome and soluble NSF attachment protein receptor (SNARE) interaction-related proteins were enriched in leaves. Using label-free quantitation, 129 differentially expressed membrane proteins were found in both tissues upon NaCl treatment. Additionally, the 140 NaCl-induced proteins identified in roots and 57 in leaves are vesicle-, mitochondrial-, and chloroplast-associated membrane proteins and those with functions related to ion transport, protein transport, ATP hydrolysis, protein folding, and receptor kinases, etc. Our proteomic results were verified against corresponding gene expression patterns from published C08 RNA-seq data, demonstrating the importance of solute transport and sensing in salt stress responses.

## 1. Introduction

Soil salinity is one of the most important environmental factors limiting plant growth and productivity throughout the world [1]. Excessive sodium in the soil inhibits the absorption of mineral nutrients and moisture, leading to the accumulation of toxic ions in plants. Plants employ several strategies to cope with salt stress at the cellular and subcellular levels. Cellular responses in plants to salinity require a new state of cellular homeostasis. These include regulating the expression of specific proteins for the reestablishment of proper cellular ion and osmotic homeostasis with other concomitant processes of repair and detoxification [2]. The salt signal is primarily perceived through roots, which rapidly respond to maintain function and transmit signals to the shoot for appropriate changes in function [3]. The regulation and integration of these cellular processes, however, are still poorly understood.

Membrane proteins are attractive targets for functional genomic studies mostly due to their important functions such as the transport of ions, nutrients, metabolites, and signaling molecules, signal transduction, bioenergetic processes, cell adhesion, and intracellular communications [4]. Due to their low abundance, poor trypsin digestibility, and instability outside of the lipid bilayers, membrane proteins are challenging to study using biochemical methods [5,6]. Thus, membrane proteome characterization has been limited. Over the past ten years, there have been many improvements in membrane proteomic methods. The development of mass spectrometry (MS)-compatible detergents, filter-assisted sample preparation, surface capture protocols, and significant advancements in MS instrumentation have significantly improved the understanding of membrane proteomes [4].

MS-based proteomics has become a standard analytical tool in various biological research fields, from basic research to membrane protein research, revealing the critical physiological roles of proteins [7]. The proteome of soybean subjected to salinity has been analyzed using roots and hypocotyls of young seedlings and other tissues, and indicated that photosynthesis, protein biosynthesis, and ATP biosynthesis-related proteins decreased while defense proteins increased in soybean in response to salt stress [8]. Further comparative proteomic approaches have been employed to explore proteome expression patterns in germinating soybeans under salt treatments, with the results suggesting that enhanced energy metabolism and accelerated protein processing in the endoplasmic reticulum were important strategies for the response to NaCl stress in germinating soybean [9]. In addition, proteomic approaches were also applied to seedlings of different salt-tolerant soybean genotypes under salt stress [10], which identified several proteins as potential candidates for augmenting salt tolerance in soybean. Several ion-exchanger genes have been identified to contribute to NaCl stress tolerance in soybean [11,12] and are the integral components of the plasma membrane. Resequencing of a recombinant inbred line population [13] led to the identification of a salt tolerance gene (*Glyma03g32900*) named *GmCHX1* which encodes an ion transporter localized on the membrane [13]. GmCAX1, a plasma membrane-localized cation/H^+^ antiporter in soybean, was reported to be induced in soybean by treatment with Na^+^ and other osmoticums such as polyethylene glycol (PEG) [14]. Several soybean calcium-dependent protein kinases, which are membrane-bound with both kinase and calcium sensor domains, were recently reported to be upregulated in response to abscisic acid (ABA) and drought treatments [15].

Most studies that investigated salt responses in soybean have focused on the relatively late responses to salinity treatment. In contrast, the early responses of plants to long-term salt stress have been overlooked [9,16]. Therefore, the main objective of this study was to investigate the membrane proteome and identify differentially expressed membrane proteins under long-term salt stress in soybean seedlings to evaluate their early responses. We used label-free quantitative proteomic techniques to detect these differentially expressed membrane proteins in leaves and roots in response to 50 mM and 100 mM NaCl treatments in the salt-sensitive genotype, C08. Our approach was sensitive enough to identify salt-induced membrane proteins with significantly altered abundance and advanced our understanding of salt-responsive mechanisms in the cell membranes of the soybean plant.

## 2. Results

### 2.1. Salt Stress Hinders Soybean Root and Shoot Growth

The growth of cultivated soybean C08 seeds germinated under salt stress at 50 and 100 mM NaCl for seven days was retarded compared to the control, with more pronounced effects exhibited in roots, resulting in a significant reduction in the numbers and lengths of lateral roots, and the effects increased with the concentration of NaCl in the treatment, although 50 mM NaCl did not induce any obvious root phenotypic differences except for reduced lateral root growth. Salt treatment also caused the unifoliate leaf margins to roll inward and hindered their expansion compared to control (Figure 1), with reduced internode, stem length, and chlorosis. The resultant phenotypes support the hypothesis that the exogenous NaCl application perhaps has an inhibitory effect on soybean seed germination and radicle length, as previously observed [17].

### 2.2. Leaf and Root Membrane Proteome under Salt Stress Revealed by Orbitrap

To probe the responses of soybean plasma membrane towards salinity and to search for clues to the mechanisms of these responses, label-free quantitative proteomics, was used to analyze the leaf and root samples from 7-day-old soybean C08 seedlings germinated under salt stress, with three biological replicates for each treatment and tissue. The identified proteins and their relative abundance are listed in Supplementary Appendix A, with an average of 1000–1200 total proteins identified from leaf and root tissues, respectively (Figure 2). Gene ontology (GO) annotations and Kyoto Encyclopedia of Genes and Genomes (KEGG) pathway analyses of the whole-proteome datasets from both tissues were compared based on the molecular functions of the identified proteins between the control and salt-treated samples and categorized by their *p*-values. Overall, the proteome datasets from both leaf and root tissues were enriched in membrane proteins involved in transporter-related activities such as active ion transmembrane transporters, ATPase-coupled transmembrane transporters, calcium ion binders, GTP binders, channel binders, monovalent transmembrane transporters, proton transmembrane transporters, soluble NSF attachment protein receptors (SNAREs), and proton transporters that are typically present in lower numbers in normal plant proteome datasets (Figure 3). This suggests our methodology is a good approach to identify transporter-related proteins in plants.

In the root, the 50 mM NaCl treatment induced many proteins, such as transmembrane transporters, pyrophosphatases, ion channels, hydrolases, and anion binding proteins, which were absent in the control dataset (Figure 3b). In addition, the 100 mM salt treatment also activated proteins such as NADPH dehydrogenases and flavoprotein oxidoreductases, which were missing in both control and 50 mM NaCl-treated root samples. It has been reported earlier that that ion channels and other transmembrane transporters are first to be activated upon salt stress, and they continue to function to mitigate more severe or prolonged salt exposure [18]. In the root, SNARE-binding activities were drastically reduced upon salt stress, which warrants further investigation into how roots cope with salt stress in soybean and other plants (Figure 3b).

The leaf proteome data revealed that salt stress induced many proteins such as cation transmembrane transporters, anion transporters, passive membrane transporters, voltage-gated ion channel proteins, superoxide dismutase copper chaperone, and translation factor binders (Figure 3a). Significant changes in many proteins related to NADPH dehydrogenases, endopeptidases, translation elongation, and monovalent inorganic cation transporters were observed in the 100 mM salt-treated leaf samples compared to control or 50 mM salt-treated samples.

KEGG pathway analyses of the leaf and root proteomes provided information on the pathways activated under salt stress. In the root, significant enrichment of proteins involved in linoleic acid metabolism pathway, endocytosis pathway, protein export pathway, and fatty acid biosynthesis pathway was observed in salt-treated samples compared to controls (Figure 3d). Overall, fatty acid metabolism, carbon metabolism, biosynthesis of amino acids, and 2-oxocarboxylic acid metabolism-related proteins were significantly enriched in salt-treated root samples (Figure 3d). In the leaves, salt stress induced different pathways, particularly those related to phagosome, spliceosome, and SNARE interactions in vesicular transport (Figure 3c). A significant enrichment in porphyrin and chlorophyll metabolism was observed in 100 mM salt-treated leaf samples compared to control and 50 mM salt-treated samples (Figure 3c).

In the identified proteomes, about 50% of the proteins in both leaf and root tissues were categorized as membrane proteins under cellular component in the UniProt Knowledgebase [19]. All the identified membrane proteins had transmembrane domains (Supplementary Appendix A). In total, we identified 972 membrane proteins from the leaf and root tissues combined, in which 421 proteins were commonly shared by both tissues, 320 membrane proteins were root-specific, and 233 were leaf-specific (Figure 4a). The 972 membrane proteins were further categorized on the basis of their subcellular locations, enzyme activities and the domains they contain (Supplementary Appendix A). Based on enzymatic functions, they can be further sorted into 158 transporter proteins, 107 oxidoreductases, 88 primary active transporters, 79 membrane traffic proteins, 41 ATP synthases, 25 SNARE proteins, and 23 chaperone proteins (Figure 4b). According to cellular component classification, there were 374 proteins identified as integral components of the membrane, 47 membrane proteins, 25 chloroplast thylakoid membrane components, 23 mitochondrial inner membrane components, 21 endoplasmic reticulum membrane components, and 20 proteins associated to the SNARE complex, photosystem I and chloroplast envelope (Figure 4c). The predominant domains in our membrane proteomes were P-loop-containing nucleoside triphosphate hydrolases, AAA^+^ ATPases, small GTPase-binding protein domains, chlorophyll a/b binding protein domains, C2 calcium-dependent membrane domains, ABC transporters, Band7 proteins, porin domain, remorins, aquaporins, syntaxins and stomatins (Figure 4d).

### 2.3. Differentially Expressed Membrane Proteins (DEMPs) in Soybean Leaves and Roots under Salt Stress

Using label-free quantification, we identified 129 differentially expressed membrane proteins (DEMPs) from both tissues and the volcano plots drawn based on a −log_10_ *p*-value > 0.05 and absolute fold-change value > 0.5 (Figure 5 and Table 1). In the leaf, 50 membrane proteins were upregulated and 21 were downregulated, while in the root, 26 were upregulated and 32 downregulated in response to salt stress. GO annotation enrichment analysis of the DEMPs revealed that the membrane proteins upregulated in the leaf were involved in nearly every aspect of transport mechanisms, purine nucleoside biosynthesis and metabolic processes, indicating the important roles of various transporters (e.g., proton transmembrane transporters, cation transporters, energy-coupled transporters, etc.) to combat salt stress in soybean (Appendix A). DEMPs related to membrane and transport were reported to be involved in early events in salt signal transduction [14]. On the other hand, the downregulated membrane proteins in leaf were mostly involved in photosynthetic processes such as light reaction, protein chromophore linkage, response to light stimulus, and generation of precursor metabolites and energy (Appendix A). All the leaf DEMPs are presented in Table 1 and Table 2 along with their gene Ensemble IDs, descriptions, Arabidopsis homologs identifier, number of peptides detected, PSMs (peptide spectrum matches), molecular weights and isoelectric points. Among the leaf DEMPs, there were several highly upregulated membrane proteins such as GLYMA_08G224400 (V-type proton ATPase catalytic subunit A), GLYMA_11G120200 (probable NAD[P]H dehydrogenase), GLYMA_11G179300 (NAD[P]H-ubiquinone oxidoreductase C1), GLYMA_02G272600 (RESTRICTED TEV MOVEMENT 2), GLYMA_13G276500 (proton pump-interactor 1), and GLYMA_09G040600 (abscisic acid receptor PYL12) (Figure 5). On the other hand, the downregulated membrane proteins in leaf contained many chloroplast membrane proteins such as GLYMA_12G059600 (plastid TRANSCRIPTIONALLY ACTIVE 16), GLYMA_05G246900 (PRA1 family protein B5), GLYMA_16G165200 and GLYMA_02G305400 (chlorophyll a/b binding proteins 1 and 1.2), and GLYMA_02G147200 (RAN GTPase-activating protein 1), suggesting a new role of chloroplast membrane proteins in salt stress signaling, or it could simply be a result of the winding down of photosynthetic activities in exchange for ramping up salt stress responses (Figure 5 and Table 1).

In the root, the 26 upregulated membrane proteins were mostly involved in the establishment of localization, nitrogen compound transport, intracellular transport, and inorganic cation and ion transmembrane transport, whereas most of the 32 downregulated membrane proteins were involved in cellular protein localization, mitochondrial transport and macromolecule localization (Figure 6, Table 3 and Table 4). The most highly upregulated proteins included GLYMA_05G215100 (COP1-interactive protein 1), GLYMA_05G078000 (wall-associated receptor kinase-like 6), GLYMA_14G126300 (cytochrome b-c1 complex subunit 7-2), GLYMA_11G146900 (receptor-like protein 4), GLYMA_08G348200 (alpha-mannosidase I MNS4), GLYMA_04G174800 (ATPase 11, plasma membrane-type), GLYMA_08G020300 (V-type proton ATPase subunit E1 [V-ATPase subunit E1]) and GLYMA_10G220100 (Ras-related protein RABG3a) (Figure 6 and Table 3). The major downregulated membrane proteins in the root were: GLYMA_10G211000 (Aquaporin PIP2-2), GLYMA_04G220400 (probable inactive receptor kinase), GLYMA_16G166600 (2-succinyl-6-hydroxy-2,4-cyclohexadiene-1-carboxylate synthase), GLYMA_13G058400 (membrane-anchored ubiquitin-fold protein 2), GLYMA_09G114500 (exocyst complex component SEC3A), GLYMA_17G021200 (probable prolyl 4-hydroxylase 4), GLYMA_04G095900 (tetratricopeptide repeat domain-containing protein PYG7), GLYMA_11G122300 (mitochondrial import receptor subunit TOM6 homolog), GLYMA_20G006500 (proton pump-interactor 1), and GLYMA_07G059400 (SUN domain-containing protein 1) (Figure 6 and Table 4).

### 2.4. Transcriptomic Verifications of DEMPs in Root and Leaf Tissues under Salt Stress

To validate the DEMPs at the transcript level, we cross-checked transcript expression patterns in a previously published dataset [20] at 24 and 48 h of salt treatment in soybean leaf and root tissues. We found that most of the DEMPs had differential gene expression patterns consistent with the differential protein expression levels in this study (Figure 7). In the leaves, 37 DEMPs had matching gene expression patterns in response to salt stress. The key upregulated proteins with doubled transcript expression levels were: GLYMA_13G088700 (annexin D1-related), GLYMA_13G295500 (solute carrier family 25 [mitochondrial adenine nucleotide translocator]), GLYMA_14G039100 (elongation factor 1-beta), GLYMA_16G026500 (uncharacterized protein), GLYMA_18G062900 (peptidyl-prolyl cis-trans isomerase), GLYMA_05G064300 (Ras-related protein Rab-1A), GLYMA_03G214500 (NADH dehydrogenase [ubiquinone) Fe-S protein 5), GLYMA_05G064300 (Ras-related protein Rab-1A [RAB1A]), GLYMA_08G224400 (V-type H^+^-transporting ATPase subunit A (ATPeV1A, ATP6A), GLYMA_13G276500 (proton pump-interactor 1), and GLYMA_10G012800 (cytochrome C oxidase subunit 5C) (Figure 7). The key downregulated DEMPs due to salt stress with matching differential gene expression levels were: GLYMA_02G305400 (light-harvesting complex II chlorophyll a/b binding protein 2), GLYMA_03G056700 (Delta [12]-fatty acid dehydrogenase), GLYMA_07G049000 (CURVATURE THYLAKOID 1B), GLYMA_08G074000 (light-harvesting complex II chlorophyll a/b binding protein 6), GLYMA_16G165200 (light-harvesting complex II chlorophyll a/b binding protein 1), GLYMA_19G007700 (carbonic anhydrase 2, chloroplastic-), and GLYMA_07G057200 (chitinase domain-containing protein 1) (Figure 8).

In the root, we found 14 upregulated transcripts in which three proteins were encoded: GLYMA_10G024000 (phospholipid: diacylglycerol acyltransferase 1), GLYMA_11G146900 (receptor-like protein 4), and GLYMA_14G126300 (cytochrome b-c1 complex subunit) having maximum gene expression at 24 h of salt treatment (Figure 8a). The other DEMPs with highly upregulated transcript levels were: GLYMA_10G107700 (calcium-transporting ATPase 4), GLYMA_06G161400 (bidirectional sugar transporter SWEET15), GLYMA_04G174800 (ATPase 11, plasma membrane-type), GLYMA_04G240400 (vacuolar protein sorting-associated protein 45 homolog), GLYMA_08G100800 (LRR receptor-like serine/threonine-protein kinase), GLYMA_08G348200 (alpha-mannosidase), and GLYMA_10G132300 (ammonium transporter 1 member 1) (Figure 8a). There were 21 DEMPs with downregulated gene expression levels at both 24 and 48 h salt treatment, including GLYMA_04G181000 (cyclic nucleotide-gated ion channel 16), GLYMA_02G307700 (fasciclin-like arabinogalactan protein 7), GLYMA_08G348000 (glucuronosyltransferase 8), GLYMA_07G059400 (SUN domain-containing protein), GLYMA_01G007700 (mitochondrial import inner membrane translocase subunit TIM17-2), and GLYMA_09G243700 (subtilisin-like protease SBT5.4) (Figure 8b). There were also some DEMPs with lowered transcript levels at 24 h salt treatment but increased expression levels at 48 h treatment, including GLYMA_05G199200 (calnexin homolog 1), GLYMA_04G220400 (probable inactive receptor kinase), GLYMA_02G086600 (binding partner of ACD11), GLYMA_10G208300 (ras-related protein RABF1), GLYMA_14G201500 (protein TIC110, chloroplastic) and GLYMA_05G016100 (CHAPERONE-LIKE PROTEIN OF POR1, chloroplastic) (Figure 8b).

### 2.5. Salt-Stress-Induced Membrane Proteins from Soybean Leaf and Root Tissues

In both root and leaf proteomes, we found 140 membrane proteins in root and 57 in leaf that were expressed only with salt treatment and absent from the control datasets based on their detection in the form of peptides. These were mainly involved in transport processes inside the cell (Appendix A). Salt-stress-induced membrane proteins in leaf were enriched in biological processes GO terms such as pyrophosphate hydrolysis-driven proton transmembrane transporter activity, protein macromolecule adaptor activity, primary active transmembrane transporter activity, inorganic molecular entity transmembrane transporter activity, ATPase-coupled cation transporter activity, porin activity, and malate antiporter activity, indicating that these membrane proteins were induced by salt stress to minimize the stress impact (Appendix A). In the root, the salt-stress-induced membrane proteins, enriched in GO terms such as cation transport, anion transport, active transmembrane transport, proton transport, electron transport, ATPase-coupled transport alongside calcium binders, clathrin heavy chain binders, oxidoreductases, and phospholipase activators, served to mitigate salt stress (Appendix A). Among the 57 potential salt-inducible membrane proteins in the leaf were GLYMA_05G019400, (ABC transporter B family member 27), GLYMA_12G172500 (aquaporin PIP2-2), GLYMA_09G056300 (ATPase 2, plasma membrane-type), GLYMA_04G203800 (bidirectional sugar transporter SWEET15), GLYMA_05G032500 (GDP-mannose transporter), GLYMA_14G099800 (mitochondrial carrier protein), GLYMA_18G297700 (piezo-type mechanosensitive ion channel), GLYMA_13G307600 (syntaxin-131), GLYMA_07G028500 (pyrophosphate-energized vacuolar membrane proton pump 1), GLYMA_20G014300 (auxin efflux carrier component 3), GLYMA_09G009300 (mitochondrial pyruvate carrier 4), and GLYMA_04G246100 (sucrose transport protein SUC9), as well as fasciclin-like arabinogalactan proteins, GLYMA_17G030700 (COBRA-like protein 7), GLYMA_09G114000 (outer envelope pore protein 24B, chloroplastic), GLYMA_08G288900 (SUPPRESSOR OF QUENCHING 1, chloroplastic), and GLYMA_14G035500 (vesicle-associated membrane protein 713), suggesting that salt stress impacts multiple processes inside the cell (Table 5).

In the root, there were multiple transporter proteins that were induced by salt treatment, including GLYMA_13G220000 (ABC transporter C family member 2), GLYMA_10G167800 (ammonium transporter 1 member 2), GLYMA_17G124900 (high-affinity nitrate transporter 3.1), GLYMA_10G057000 (membrane magnesium transporter), GLYMA_16G046200 (potassium transporter 4), GLYMA_08G092000 (sodium/hydrogen exchanger 7), and GLYMA_08G180400 (sulfate transporter 1.3), showing the important role of transporter proteins in response to salt stress. There were also membrane receptor kinases induced by salt stress, such as GLYMA_08G180800 (BRI1-associated receptor kinase 1), GLYMA_09G110500 (L-type lectin-domain containing receptor kinase S.5), and GLYMA_12G212200 (Serine/threonine-protein kinase VPS15) (Table 6). Phospholipase-related membrane proteins were particularly abundant in root tissues upon salt stress treatment, including GLYMA_11G085100 (phosphoinositide phosphatase SAC7), GLYMA_10G183200 (phosphatidate cytidylyltransferase 2), GLYMA_07G031100 (phospholipase D alpha 1), and GLYMA_14G059200 (phosphoinositide phospholipase C2) (Table 6). There were also vesicle-associated, vacuolar, chloroplastic and mitochondrial membrane proteins induced by salt tress, including GLYMA_06G067800 (vesicle putative clathrin assembly protein), GLYMA_12G032400 (synaptotagmin-2 vesicle), GLYMA_07G119500 (vacuolar protein-sorting-associated protein 11 homolog), GLYMA_06G023000 (vacuolar Protein FREE1), GLYMA_18G217200 (RETICULATA-RELATED 3, chloroplastic), GLYMA_06G039500 (TIC 22, chloroplastic), GLYMA_12G094000 (alternative NAD[P]H-ubiquinone oxidoreductase C1, chloroplastic/mitochondrial), GLYMA_06G019400 (ATP-dependent zinc metalloprotease FTSH 5, chloroplastic), GLYMA_03G114600 (oxygen-evolving enhancer protein 3-2, chloroplastic), GLYMA_05G110100 (cytochrome c oxidase subunit 6a, mitochondrial), GLYMA_10G286200 (dihydroorotate dehydrogenase [quinone], mitochondrial), GLYMA_10G049900 (mitochondrial import inner membrane translocase subunit), and GLYMA_07G193700 (mitochondrial inner membrane protein OXA1). Membrane-trafficking proteins such as GLYMA_12G001900 (patellin-3), GLYMA_14G199000 (sorting nexin 2B), and vacuolar proton pump proteins GLYMA_15G044700 (v-type proton ATPase subunit a1), GLYMA_20G106500 (v-type proton ATPase subunit B2) and GLYMA_02G065500 (v-type proton ATPase subunit C), and trafficking signal-inducing ethylene response protein GLYMA_16G071500 (EIN2-CEND) were also induced by salt stress in soybean root tissues (Table 6).

## 3. Discussion

To cope with salt stress, soybean plants have evolved complex salt-responsive signaling at the membrane, cellular, organ, and whole-plant levels. Membrane proteins play a variety of roles in crucial biological functions, for example, the transport of biological substances (ions, nutrients, metabolites and signaling molecules), signal transduction, bioenergetic processes, immune response, cell adhesion, and cell–cell interactions. These proteins are challenging to study using biochemical methods because they are only present at low levels and are unstable outside of the lipid bilayers [5]. Hence, they are seldom studied despite their significance in biological systems. Nonetheless, the growing demand for increased quality and quantity of membrane proteomic data has resulted in improved membrane proteomics methodology. High-throughput MS has contributed significantly to the comprehensiveness of proteomics, resulting in the identification of many membrane proteins [4]. Here, we carried out unbiased label-free quantitative proteomic analyses comparing the relative abundance of membrane proteins in salt-treated samples of soybean leaves and roots compared to controls. Among the 972 membrane proteins identified, a total of 233 proteins in leaves and 320 proteins in roots responded to NaCl-stress treatments.

The possible observed pictorial phenotypes of decreased plant height, reduced internode and lateral root number and length were similar to those reported in previous soybean studies where exogenous NaCl applications had a remarkable inhibitory effect on plant growth [21,22] (Figure 1).

Our detected membrane proteome included proteins such as active ion transmembrane transporters [23], ATPase-coupled transmembrane transporters [24], calcium ion binders [25], GTP binders [26], channel proteins [12], monovalent transmembrane transporters [18], transmembrane proton transporters [27], and SNARE receptors [28], confirming their involvement in salt stress tolerance observed in previous studies. In the root, upon 100 mM salt treatment, the activation of NADPH dehydrogenases and flavoprotein oxidoreductases clearly indicated that NADPH regeneration would be essential in the defense mechanism against salt-induced oxidative stress [29].

In leaves, the enrichment of KEGG pathways involved in phagosome, spliceosome and SNARE interactions in vesicular transport strongly suggested that certain splicing factors and autophagy could increase the plant tolerance to salt stress [30,31] (Figure 3c). In the root, the enrichment of KEGG pathways involved in linoleic acid metabolism and endocytosis suggested that salt-stress responses such as lipid metabolism and increased plasma membrane endocytosis could be possible mechanisms for mitigating salt stress in plants [32,33] (Figure 3d). Conserved membrane proteins, including Band7 proteins, porins, remorins, aquaporins, syntaxins and stomatins, could be the major responsive proteins against salt stress in plants because they are crucial in transport, signaling, bioenergetics, and catalysis [34,35,36].

In the leaf, the upregulated proteins, such as abscisic acid receptor PYL12 (GLYMA_09G040600), COP1-interactive protein 1 (GLYMA_11G096400), two cation/H(+) antiporter 8 (GLYMA_09G222500; GLYMA_12G036000), Synaptotagmin-1 (GLYMA_11G107300), V-ATPase subunit A, V-ATPase subunit E3, E1 (GLYMA_08G224400, GLYMA_05G214200, GLYMA_08G020300), and aquaporin NIP2-1 (GLYMA_12G066300), suggest their novel roles in salt-stress responses towards sodium toxicity and ABA-mediated activities inside leaf cells (Table 1 and Figure 4). A soybean sodium/hydrogen exchanger enhanced the salt tolerance through maintaining a higher Na^+^ efflux rate and K^+^/Na^+^ ratio in Arabidopsis [37]. In rice, a Na^+^/H^+^ exchanger protein is the sole Na^+^ efflux transporter and its loss-of-function mutant displayed exceptional salt sensitivity [38]. In Arabidopsis, *synaptotagmin-1* mutant plants were severely affected by salt stress since this protein is required for the maintenance of plasma membrane integrity [39,40]. During salinity stress, V-ATPases facilitate the sequestration of Na^+^ in the vacuole by establishing an electrochemical proton gradient across the tonoplast [41], and their abundances and activities are usually increased during salinity stress. An adaptive response and induced expression of Nodulin 26-like Intrinsic Protein 2-1 (NIP2-1) were recorded in Arabidopsis under low-oxygen stress [42].

Many proteins related to chloroplasts and photosynthesis were downregulated upon salt stress treatment in soybean leaves, including chlorophyll a/b binding protein 2.1 (GLYMA_02G305400), ATP-dependent zinc metalloprotease FTSH 5 (GLYMA_04G019100), CURVATURE THYLAKOID 1B (GLYMA_07G049000), Photosystem I chlorophyll a/b-binding protein 5 (GLYMA_08G074000), Photosystem I reaction center subunit N (GLYMA_10G256000), PLASTID TRANSCRIPTIONALLY ACTIVE 16 (GLYMA_11G135900), Photosystem II protein D1(GLYMA_13G028200), and Beta-carbonic anhydrase 1, chloroplast (GLYMA_19G007700), indicating that salinity-stress-enhanced reactive oxygen species (ROS) production led to severe damage to chloroplasts (Table 2 and Figure 4). The lowered expressions of chloroplast- and photosynthesis-related genes under salt stress have also been reported previously in soybean leaves [20,43].

In the root, differentially expressed ion pump proteins, which included vacuolar-type proton ATPase subunit E1 (GLYMA_08G020300), ATPase 11, plasma membrane-type (GLYMA_04G174800), and calcium-transporting ATPase 4, plasma membrane-type (GLYMA_10G107700), were found to be critical for plant salt tolerance and had key roles in regulating ion transport under salt stress (Table 3 and Figure 5). An improved method using tandem affinity purification tag for the salt-overly sensitive (SOS) pathway indicated that subunits A, B, C, E and G of the peripheral cytoplasmic domain of the vacuolar ATPase were present in an SOS2-containing protein complex to maintain favorable ion ratios in the cytoplasm and for increasing the tolerance of salt stress [9,44]. An overexpressing form of the plasma membrane ATPase that is constitutively active conferred increased salt tolerance [45,46]. It has been reported that the salt-stress-induced elevation in cytosolic Ca^2+^ and the new cytosolic Ca^2+^ status is regulated by Ca^2+^-ATPases [47]. In the root, the transporter proteins that were highly upregulated by salt stress included ammonium transporter 1 (GLYMA_10G132300) and high affinity nitrate transporter 2.5 (GLYMA_18G141900), suggesting that ammonium transport could alleviate ammonia toxicity caused by salt stress. The overexpression of *PutAMT1;1* gene promoted early root growth after seed germination in transgenic Arabidopsis under salt stress [48]. In salt-stressed roots, the upregulation of serine/threonine-protein kinase PBL19 (GLYMA_01G181700), wall-associated receptor kinase-like 6 (GLYMA_05G078000), LRR receptor-like serine/threonine-protein kinase (GLYMA_08G100800), receptor-like protein 4 (GLYMA_11G146900), and calcium-dependent protein kinase 2 (GLYMA_15G222300) indicated that they are potential salt-stress sensors in the root (Table 3). SlWAK1 is a wall-associated kinase involved in salt tolerance in tomato (*Solanum lycopersicum*). *slwak1* mutants are tolerant to Na^+^ stress, but not to osmotic stress [49]. For example, in *Medicago truncatula*, the LRR-RLK gene, *Srlk*, was rapidly induced in roots in response to salt stress [6]. In potato, calcium-dependent protein kinase 2 was identified as a sensor-transducer in the salt stress response in potato plants. Its overexpression promoted ROS scavenging, chlorophyll stability and the induction of stress-responsive genes, thus conferring tolerance to salinity [50].

In the root, the downregulated membrane proteins upon salt stress included aquaporin PIP2-2 (GLYMA_10G211000), proton pump-interactor 1 (GLYMA_20G006500), calnexin homolog (GLYMA_05G199200), (GLYMA_09G243700), and ras-related protein RABF1 (GLYMA_10G208300) (Table 4 and Figure 5). In Arabidopsis, salt stress is known to localize PIP2 to intracellular compartments, probably to decrease the water permeability of the root [51]. In Arabidopsis, subtilase 6.1 associated with the unfolded protein responded to salt stress through the cleavage of an ER-resident type II membrane protein (bZIP28) [52]. In tobacco, the calreticulin gene from wheat (*TaCRT1*) improved the salinity tolerance [53]. In Arabidopsis, RabF1 is involved in salt-stress responses through endosomal vesicle transport and may play a crucial role in the recycling and degradation of molecules [54]. We found some chloroplast-related proteins downregulated under salt stress in roots, such as GLYMA_04G095900 (tetratricopeptide repeat domain-containing protein PYG7), GLYMA_05G016100 (CHAPERONE-LIKE PROTEIN OF POR1), GLYMA_14G076000 (outer envelope pore protein 24B), and GLYMA_14G201500 (TIC110), which are essential for chloroplast development, and their suppression plays a role in salt stress responsiveness in chloroplasts. TIC110 is a component of the protein import apparatus in the chloroplast, and its downregulation suggests that salt stress hinders the process of importing nuclear DNA-encoded proteins into chloroplasts. In Arabidopsis, the downregulation of TIC110 expression resulted in the reduced accumulation of a wide variety of plastid proteins [55]. The downregulation of mitochondrial import inner membrane translocase subunit TIM17-2 and mitochondrial import receptor subunit TOM6 indicates that salt stress hinders the import processes in mitochondria in response to stress [56]. The downregulation of mitochondrial ADP, ATP carrier protein 3 (GLYMA_12G205400), means that salt stress disrupts the supply of ATP from the mitochondrial matrix to the cytosol [57].

We also found some proteins in both soybean leaf and root that were present only in the salt-treated samples, suggesting that they were only produced in response to salt. In the root, salt stress induced many transporter proteins, such as ABC transporter C and G family members, membrane magnesium transporter, potassium transporter 4, sodium/hydrogen exchanger 7 (SOS1), and sulfate transporter 1.3 (Table 5 and Table 6). Under salinity stress, the ABC transporters have been reported to enhance plant tolerance through the sequestration of sodium salt [58]. In rice, a Mg transporter, OsMGT1, is required for salt tolerance probably by regulating the transport activity of OsHKT1;5, a key transporter for the removal of Na^+^ from the xylem sap at the root maturation zone [59]. In soybean, salt treatment elevated the expressions of *SOS1* and *AKT1* genes, and reduced the expressions of *SKOR* and *HKT1* genes [37]. A significant accumulation of Na^+^ in the roots of the *gmsos1* mutants resulted in the imbalance between Na^+^ and K^+^, which impaired Na^+^ efflux and increased K^+^ efflux in the roots of soybean under salt stress [60]. Interestingly, the expression of the sulfur transporter *AtSULTR3;1* was upregulated in response to drought and salt stress [61].

## 4. Materials and Methods

### 4.1. Soybean Stress Treatment

The surface-sterilized seeds of *G. max* (accession C08) were germinated in moist vermiculite supplemented with 50 mM or 100 mM NaCl treatments in the greenhouse under normal conditions. After 7 days, whole plant roots and the unifoliate leaves were collected for membrane proteome extraction and quantification. Three independent sets of control and NaCl-treated samples were collected, and each biological replicate represented a pooled sample of three individual plants.

### 4.2. Extraction of Membrane Proteins and Trypsin Digestion

Liquid-nitrogen-frozen leaf and root tissues (about 100 mg) were ground to a fine powder with a pestle and mortar, and membrane proteins were then extracted using a membrane extraction kit according to the manufacturer’s protocol using plasma membrane extraction protocols (Cat#SM-005-P; Invent Biotechnologies, Plymouth, MN, USA). The extracted proteins were dissolved in 8 M urea buffer for further digestion and sample preparation.

Digestion of proteins was performed using SMART digest^TM^ trypsin kit (60109-101; Thermo Scientific, Waltham, MA, USA) in solution. Protein reduction and alkylation were achieved with 10 mM dithiothreitol at 56 °C for 30 min followed by 25 mM iodoacetamide at room temperature for 25 min. The digested peptides were purified using Pierce^TM^ C-18 spin columns (Thermo Scientific, Waltham, MA, USA) and finally dissolved in 0.1% formic acid (FA).

### 4.3. Orbitrap-Based Liquid Chromatography-Tandem MS (LC-MS/MS) Analysis

MS analysis was performed using an Orbitrap Fusion^TM^ Lumos^TM^ Tribrid^TM^ Mass Spectrometer (Thermo Scientific, Waltham, MA, USA) interfaced with an LC UltiMate 3000 RSLCnano system (Thermo Scientific, Waltham, MA, USA). The peptide separation was carried out at 50 °C with a C-18 μ-precolumn (300 μm i.d. × 5 mm) followed by Acclaim^TM^ PepMap^TM^ RSLC, on a nanoViper C-18 column, 75 μm × 25 cm (Thermo Scientific, Waltham, MA, USA) at a flow rate of 0.3 μLmin^−1^, using mobile phase A (98% H_2_O, 1.9% acetonitrile (ACN) with 0.1% FA) and mobile phase B (98% ACN, 1.9% H_2_O with 0.1% FA). The following LC gradient was used to detect membrane proteins: 100% mobile phase A for the initial 5 min, followed by 0–6% mobile phase B for 3 min, 6–30% mobile phase B for 42 min, 30–45% mobile phase B for 10 min, 45–60% mobile phase B for 10 min, 60–80% mobile phase B for 5 min and an additional 5 min at 80% mobile phase B, followed by a final re-equilibrium with 100% mobile phase A for 10 min.

The Orbitrap was set up in a data-dependent MS/MS mode under direct control of the Xcalibur software (version 4.1), where a full-scan spectrum (from 375 to 1500 m/z) was followed by tandem mass spectra (MS/MS). The instrument was operated in positive mode with a spray voltage of 2 kV, a capillary temperature of 300 °C, and was calibrated before measurements. Full scans were performed in the Orbitrap with a resolution of 60,000 at 400 m/z, with a precursor ion selection (AGC > 4.0e5) and ion charge >1. Higher energy collisional dissociation, performed at the far side of the C-trap, was chosen as the fragmentation method, by applying a 30% value for normalized collision energy, an isolation window of m/z 1.6, with a maximum injection time of 250 milli seconds (ms) and Orbitrap resolution of 15,000.

### 4.4. Data Analysis and Interpretation

Proteome Discoverer (version 2.3; Thermo Scientific, Waltham, MA, USA), interfaced with an in-house SEQUEST server, was used for data processing, peptide identification, and protein inference, according to the following criteria: Ensembl database of *G. max*, Enzyme Trypsin, Maximum Missed Cleavage Sites: 2, Precursor Mass Tolerance: 10 ppm, Fragment Mass Tolerance: 0.2 Da, Cysteine Carbamidomethylation as static modification, N-terminal protein acetylation and methionine oxidation as dynamic modifications. The Percolator algorithm was used for peptide validation (peptide confidence: q-value < 0.01, corresponding to false discovery rate (FDR) < 0.01) and only Rank 1 peptides were considered. Peptide and protein grouping were performed using the strict maximum parsimony principle. Label-free quantification analyses using the Minora algorithm were performed with three biological replicates for each treatment throughout the whole proteome analyses. To be considered significantly differentially expressed, a protein had to be quantified with at least three peptides in each biological replicate, a *p*-value <0.05, and a fold-change > 0.5 for upregulated protein or <−0.5 for down-regulated proteins. The functional annotation of proteins was determined by The Database for Annotation, Visualization and Integrated Discovery (DAVID) and ShinyGo, and then grouped on the basis of their molecular functions, cellular components, and biological processes from Gene Ontology (GO) terms combined with information from the literature [3,26]. Domain analyses were performed using the InterPro website. Volcano plots of differentially expressed proteins were drawn using VolcaNoseR (https://huygens.science.uva.nl/VolcaNoseR/, accessed on 22 February 2022) and by plotting log_10_ *p*-values on the Y-Axis against the log_2_ fold-changes on the X-axis. The corresponding RNAseq-based gene expression values of significantly differentially expressed proteins were retrieved from [20] at 24 h and 48 h of salt treatment, and heatmaps were drawn using the heatmapper website.

## Figures and Tables

**Figure 1 ijms-23-13270-f001:**
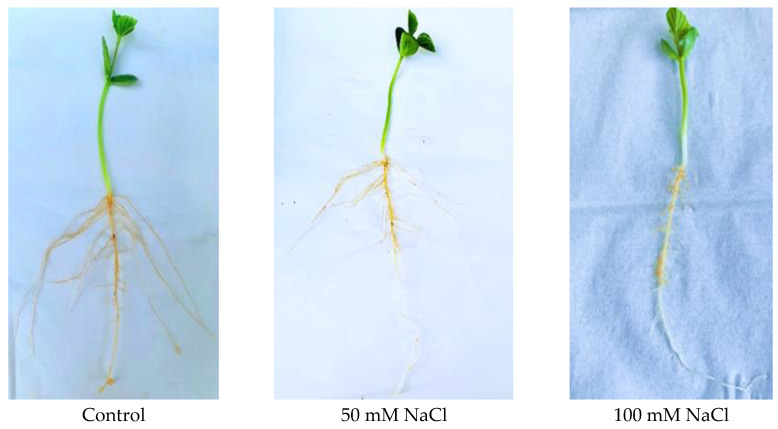
Phenotypes of salt-germinated 7-day-old seedlings of the cultivated soybean genotype C08 under 50 mM and 100 mM NaCl. Treated plants showed an overall stunted growth with decreased root and shoot growth.

**Figure 2 ijms-23-13270-f002:**
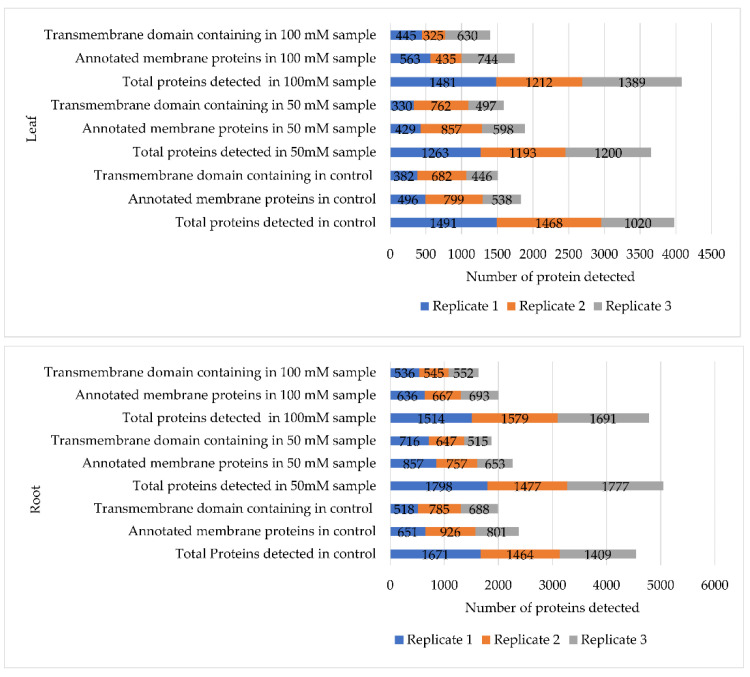
Comparisons of the numbers of total identified proteins against the numbers of transmembrane domain-containing proteins among all three biological replicates in both leaf (upper panel) and root tissues (lower panel).

**Figure 3 ijms-23-13270-f003:**
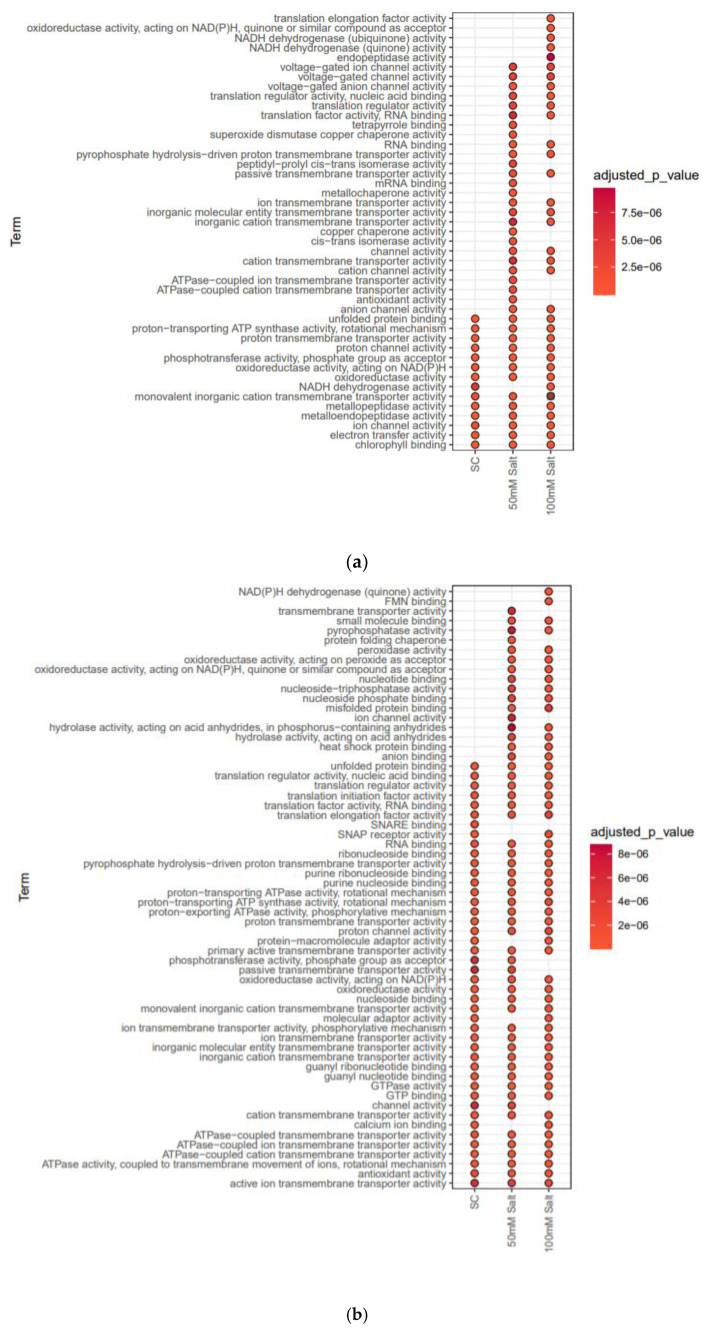
Gene ontology (GO) analyses and Kyoto Encyclopedia of Genes and Genomes (KEGG) pathway analyses of whole-proteome data of leaf and root tissues under control (SC) and 50 mM and 100 mM NaCl treatments. (**a**) Molecular-function GO enrichment of identified proteins in leaves under control and salt treatments. (**b**) Molecular-function GO enrichment of identified proteins in roots under control and salt treatments. (**c**) KEGG pathway enrichments of control and salt-treated leaf samples. (**d**) KEGG pathway enrichments of control and salt-treated root samples.

**Figure 4 ijms-23-13270-f004:**
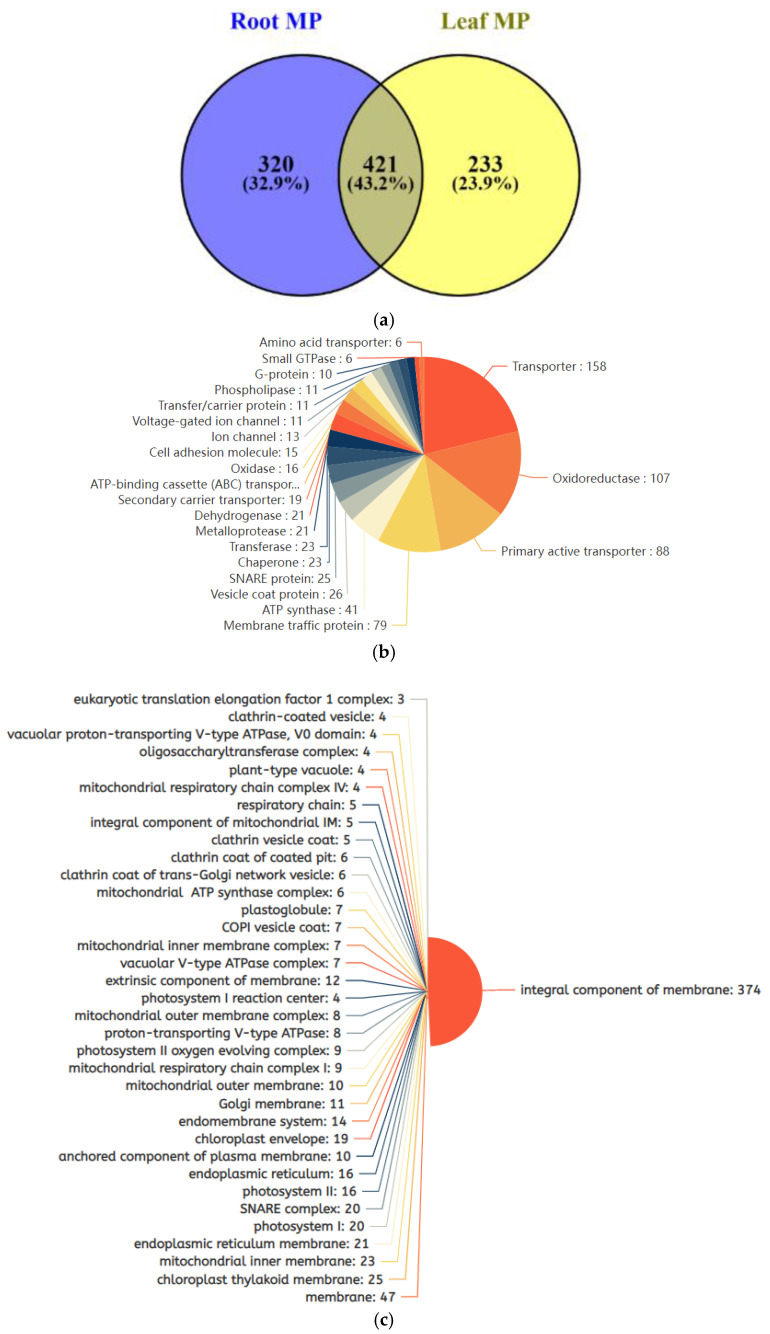
Overview of identified transmembrane domain-containing proteins from both leaf and root tissues in various categories based on their molecular functions, cellular components and types of proteins. (**a**) Venn diagram showing the numbers of commonly shared, leaf-specific and root-specific membrane proteins. (**b**) Distributions of membrane proteins according to molecular functions. (**c**) Distributions of the membrane proteins in various cellular components. (**d**) Classification of the membrane proteins based on their transmembrane domains.

**Figure 5 ijms-23-13270-f005:**
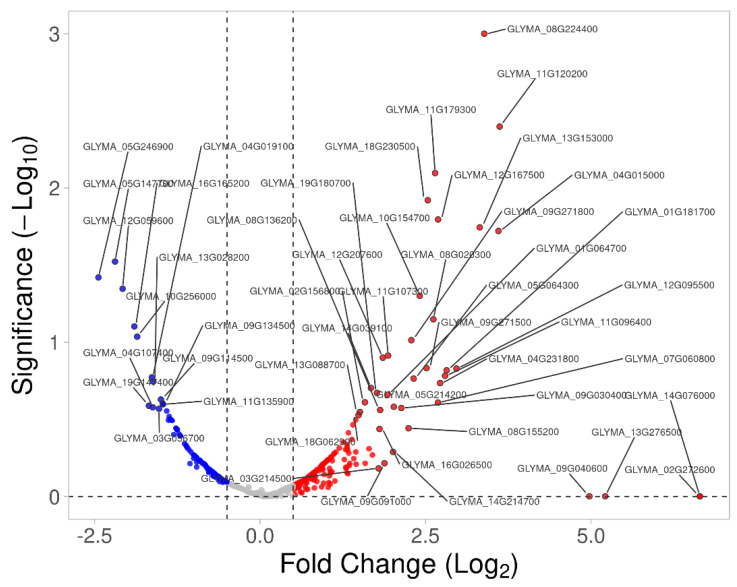
Up- and down-regulated membrane proteins in leaves identified using label-free quantification, with a −log_10_ *p*-value > 0.05 and |fold-change| value > 0.5. Red dots represent up-regulated proteins and blue dots represent down-regulated proteins. Gray dots represent membrane proteins with unchanged expressions.

**Figure 6 ijms-23-13270-f006:**
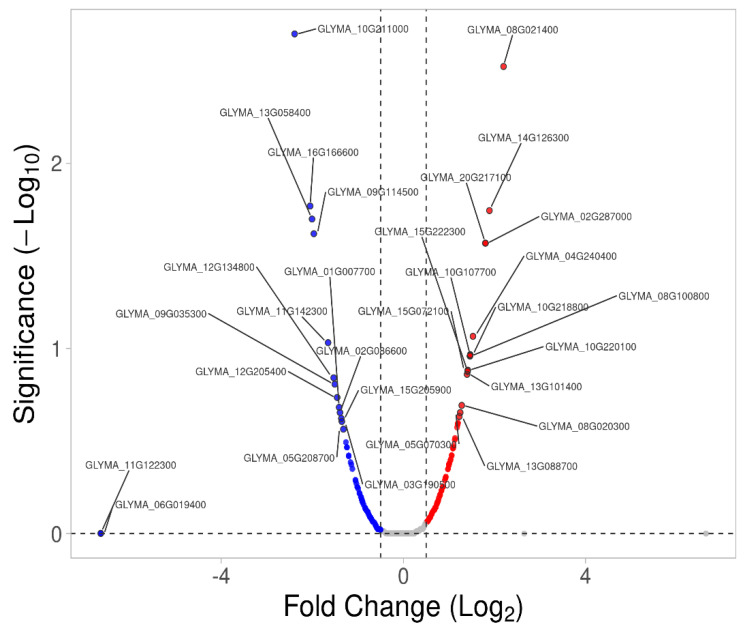
Up- and down-regulated membrane proteins in roots identified using label-free quantification, with a −log_10_ *p*-value > 0.05 and |fold change| value > 0.5. Red dots represent up-regulated proteins and blue dots represent down-regulated proteins. Gray dots represent membrane proteins with unchanged expressions.

**Figure 7 ijms-23-13270-f007:**
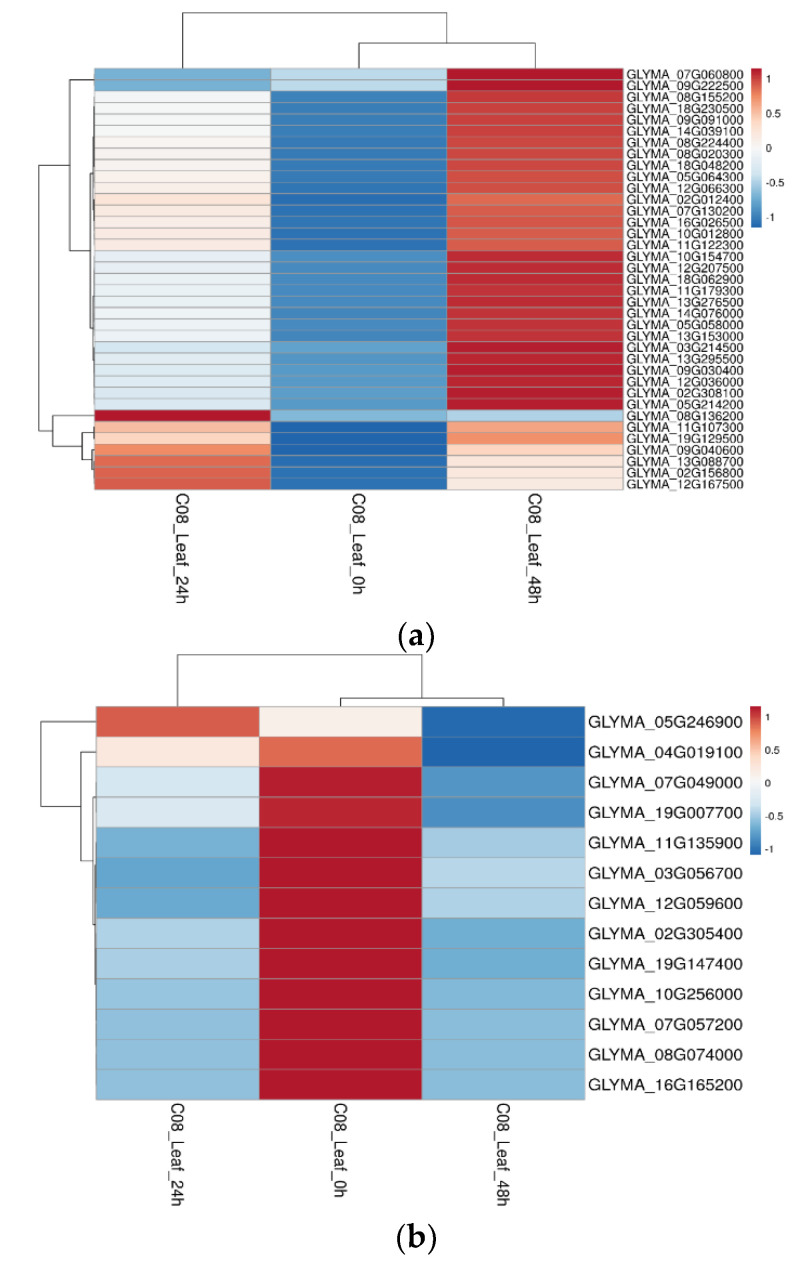
Gene expressions of up- and down-regulated membrane proteins in the leaf of soybean C08 upon salt treatment were consistent with the corresponding protein expression patterns. (**a**) Heat map of the gene expressions of key up-regulated membrane proteins in the leaf based on RPKM (reads per kilobase million mapped reads) values extracted from the published C08 transcriptomic data on the effects of 24 or 48 h of 150mM NaCl treatment [20]. (**b**) Heat map of the gene expressions of down-regulated membrane proteins in the leaf based on RPKM values extracted from the published C08 salt treatment transcriptomic data [20]. Red bars represent up-regulated gene expressions and blue bars represent down-regulated gene expressions of the corresponding differentially expressed proteins from this study. White bars represent unchanged gene expression levels.

**Figure 8 ijms-23-13270-f008:**
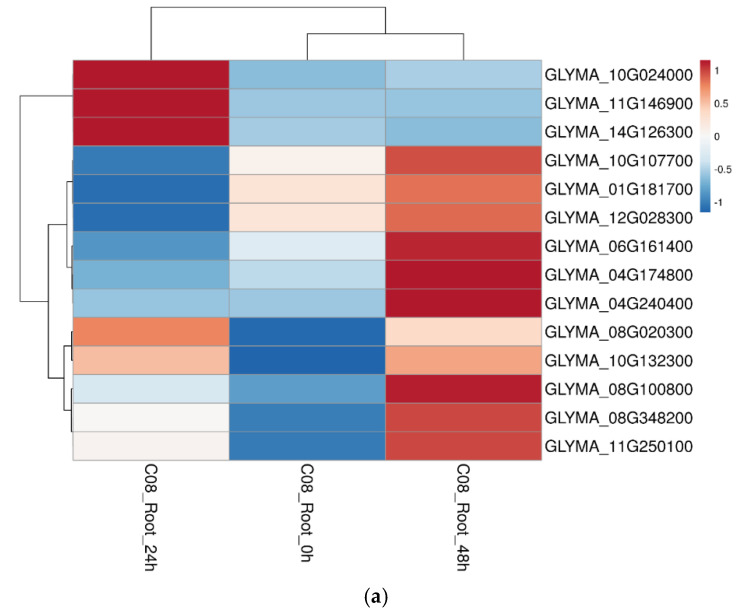
Gene expressions of up- and down-regulated membrane proteins in the root of soybean cultivar C08 upon salt treatment were consistent with the protein expression patterns. (**a**) Heat map of the gene expression levels of key up-regulated membrane proteins in the root based on RPKM values extracted from the published C08 transcriptomic data on the effects of 24 or 48 h of 150 mM NaCl treatment [20]. (**b**) Heat map of the gene expression levels of key down-regulated membrane proteins in the root based on RPKM values extracted from the published C08 salt treatment transcriptomic data [20]. Red bars represent up-regulated gene expressions and blue bars represent down-regulated gene expressions of the corresponding differentially expressed proteins from this study. White bars represent unchanged gene expression levels.

**Table 1 ijms-23-13270-t001:** Up-regulated membrane proteins under salt stress in soybean leaves.

Ensemble Protein ID	Primary Protein Name	Arabidopsis Identifier	No. of Peptides	No. of PSMs	MW [kDa]	Calc. pI
GLYMA_09G040600	Abscisic acid receptor PYL12	PYL12_ARATH	12	39	16.5	4.83
GLYMA_11G096400	COP1-interactive protein 1	CIP1_ARATH	20	58	153.2	4.7
GLYMA_09G222500	Cation/H(+) antiporter 8	CHX8_ARATH	2	12	5.7	10.17
GLYMA_12G036000	Cation/H(+) antiporter 8	CHX8_ARATH	2	10	5.6	10.17
GLYMA_08G155200	ATP synthase subunit O, mitochondrial	ATPO_ARATH	15	80	26.9	9.52
GLYMA_09G091000	NADH dehydrogenase [ubiquinone] 1 alpha subcomplex subunit 6	NDUA6_ARATH	6	45	17.1	7.81
GLYMA_05G058000	NADH dehydrogenase [ubiquinone] 1 beta subcomplex subunit 8, mitochondrial	NDUB8_ARATH	3	9	13.5	7.97
GLYMA_11G107300	Synaptotagmin-1	SYT1_ARATH	5	8	61.7	6.73
GLYMA_05G064300	Ras-related protein RABD2c (AtRABD2c)	RAD2C_ARATH	9	30	22.6	5.71
GLYMA_02G308100	Transmembrane emp24 domain-containing protein p24beta2	P24B2_ARATH	2	7	25.4	6.47
GLYMA_09G030400	Coatomer subunit epsilon-1	COPE1_ARATH	1	4	32.3	5.57
GLYMA_08G224400	V-type proton ATPase catalytic subunit A (V-ATPase subunit A)	VATA_ARATH	34	276	68.7	5.67
GLYMA_05G214200	V-type proton ATPase subunit E3 (V-ATPase subunit E3)	VATE3_ARATH	22	123	27.6	6.8
GLYMA_08G020300	V-type proton ATPase subunit E1 (V-ATPase subunit E1)	VATE1_ARATH	21	141	26.8	6.25
GLYMA_14G076000	Outer envelope pore protein 24B, chloroplastic	OP24B_ARATH	14	83	23.7	9.26
GLYMA_14G214700	Probable inactive receptor kinase At1g48480	Y1848_ARATH	2	2	68.1	7.2
GLYMA_09G271500	Probable receptor-like protein kinase At5g24010	Y5241_ARATH	3	13	35.4	7.84
GLYMA_12G167500	Uncharacterized protein At5g49945	Y5994_ARATH	7	17	50.8	6.34
GLYMA_08G136200	Annexin D2	ANXD2_ARATH	14	31	35.6	6.46
GLYMA_10G154700	Mitochondrial Rho GTPase 1 (AtMIRO1)	MIRO1_ARATH	1	4	71.9	5.59
GLYMA_11G120200	Probable NAD(P)H dehydrogenase (quinone) FQR1-like 2	FQRL2_ARATH	9	37	27.3	6
GLYMA_02G028200	Metalloendoproteinase 5-MMP (At5-MMP)	5MMP_ARATH	2	3	43.8	9.09
GLYMA_13G276500	Proton pump-interactor 1	PPI1_ARATH	36	173	69.3	9
GLYMA_12G207500	Prohibitin-2, mitochondrial (Atphb2)	PHB2_ARATH	20	105	37.1	9.63
GLYMA_18G230500	Uncharacterized protein At3g61260	Y3126_ARATH	12	40	20.1	8.7
GLYMA_09G271800	Protein RETICULATA-RELATED 3, chloroplastic	RER3_ARATH	4	4	36.9	8.27
GLYMA_12G207600	Fasciclin-like arabinogalactan protein 9	FLA9_ARATH	17	159	26.3	9.11
GLYMA_19G180700	Fasciclin-like arabinogalactan protein 11	FLA11_ARATH	1	6	26.3	5.85
GLYMA_02G012400	Cytochrome c oxidase subunit 5C-2	CX5C2_ARATH	7	41	7.1	8.59
GLYMA_10G012800	Cytochrome c oxidase subunit 5C-2	CX5C2_ARATH	7	47	7.1	8.59
GLYMA_01G064700	Protein trichome birefringence-like 37	TBL37_ARATH	1	9	21.8	6.64
GLYMA_11G179300	Alternative NAD(P)H-ubiquinone oxidoreductase C1, chloroplastic/mitochondrial	NDC1_ARATH	31	238	39.3	9.42
GLYMA_18G062900	Peptidyl-prolyl cis-trans isomerase CYP19-4 (PPIase CYP19-4)	CP19D_ARATH	7	28	21.9	9.17
GLYMA_19G129500	Rhomboid-like protein 20 (AtRBL20)	RBL20_ARATH	2	6	36.6	10.02
GLYMA_15G016300	Photosystem I reaction center subunit VI-2, chloroplastic	PSAH2_ARATH	6	28	14.9	9.99
GLYMA_01G181700	Probable serine/threonine-protein kinase PBL19	PBL19_ARATH	1	1	41.9	8.73
GLYMA_02G272600	Protein RESTRICTED TEV MOVEMENT 2	RTM2_ARATH	7	21	28.9	8.02
GLYMA_14G039100	Elongation factor 1-beta 1 (EF-1-beta 1)	EF1B1_ARATH	7	15	24.2	4.68
GLYMA_13G153000	Ras-related protein RABF2a (AtRABF2a)	RAF2A_ARATH	3	13	25.6	7.3
GLYMA_18G048200	Protein TIC 40, chloroplastic	TIC40_ARATH	8	26	47.3	6.44
GLYMA_11G122300	Mitochondrial import receptor subunit TOM6 homolog	TOM6_ARATH	2	4	6.2	9.61
GLYMA_13G088700	Annexin D1	ANXD1_ARATH	13	52	35.9	6.96
GLYMA_07G060800	Sugar transport protein 7	STP7_ARATH	4	12	27.2	10.08
GLYMA_04G015000	Monosaccharide-sensing protein 2	MSSP2_ARATH	2	7	78.7	5.72
GLYMA_07G130200	Protein DETOXIFICATION 50 (AtDTX50)	DTX50_ARATH	3	16	8.7	8.53
GLYMA_13G295500	ADP, ATP carrier protein 3, mitochondrial	ADT3_ARATH	21	129	39.6	9.85
GLYMA_12G066300	Aquaporin NIP2-1	NIP21_ARATH	6	29	7.8	10.61
GLYMA_12G095500	Putative purine permease 20 (AtPUP20)	PUP20_ARATH	4	9	27	9.39
GLYMA_16G026500	Succinate dehydrogenase subunit 8, mitochondrial	SDH8_ARATH	2	7	5.7	8.62
GLYMA_05G067900	RING-H2 finger protein ATL28	ATL28_ARATH	1	1	6.4	8.69

PSM, peptide spectrum matches; pI, isoelectric point.

**Table 2 ijms-23-13270-t002:** Down-regulated membrane proteins under salt stress in soybean leaves.

Ensemble Protein IDs	Primary Protein Name	Arbidopsis Identifier	Peptides	PSMs	MW [kDa]	Calc. pI
GLYMA_02G147200	RAN GTPase-activating protein 1 (AtRanGAP1; RanGAP1)	RAGP1_ARATH	3	3	57.7	4.78
GLYMA_02G305400	Chlorophyll a-b binding protein 2.1, chloroplastic	CB21_ARATH	8	40	28.6	5.66
GLYMA_03G056700	Temperature-sensitive sn-2 acyl-lipid omega-3 desaturase (ferredoxin), chloroplastic	FAD3D_ARATH	2	8	51.3	7.71
GLYMA_03G179500	Syntaxin-71 (AtSYP71)	SYP71_ARATH	3	9	29.7	5.94
GLYMA_04G019100	ATP-dependent zinc metalloprotease FTSH 5, chloroplastic (AtFTSH5)	FTSH5_ARATH	18	90	74.1	6.19
GLYMA_04G107400	Mitochondrial import inner membrane translocase subunit TIM44-2	TI442_ARATH	1	2	54.4	8.82
GLYMA_05G147700	Non-specific lipid transfer protein GPI-anchored 1 (AtLTPG-1; Protein LTP-GPI-ANCHORED 1)	LTPG1_ARATH	2	9	19	8.24
GLYMA_05G246900	PRA1 family protein B5 (AtPRA1.B5)	PR1B5_ARATH	1	4	22.6	8.66
GLYMA_07G049000	Protein CURVATURE THYLAKOID 1B, chloroplastic	CUT1B_ARATH	1	3	18.1	8.76
GLYMA_07G057200	Rhodanese-like domain-containing protein 4, chloroplastic	STR4_ARATH	14	43	46.4	6.23
GLYMA_08G074000	Photosystem I chlorophyll a/b-binding protein 5, chloroplastic (Lhca5)	LHCA5_ARATH	2	5	27.4	7.49
GLYMA_09G114500	Exocyst complex component SEC3A (AtSec3a)	SEC3A_ARATH	1	2	101	5.87
GLYMA_09G134500	Protein SRC2 homolog (AtSRC2)	SRC2_ARATH	5	18	30.3	5.03
GLYMA_10G256000	Photosystem I reaction center subunit N, chloroplastic (PSI-N)	PSAN_ARATH	6	20	15.4	9.67
GLYMA_11G054900	PI-PLC X domain-containing protein At5g67130	Y5713_ARATH	7	24	46.1	7.5
GLYMA_11G135900	Protein PLASTID TRANSCRIPTIONALLY ACTIVE 16, chloroplastic (pTAC16)	PTA16_ARATH	35	136	55.5	8.65
GLYMA_12G059600	Protein PLASTID TRANSCRIPTIONALLY ACTIVE 16, chloroplastic (pTAC16)	PTA16_ARATH	32	131	55	8.03
GLYMA_13G028200	Photosystem II protein D1 (PSII D1 protein)	PSBA_ARATH	3	43	29.8	5.14
GLYMA_16G165200	Chlorophyll a-b binding protein 1, chloroplastic	CB1C_ARATH	14	61	28	5.41
GLYMA_19G007700	Beta carbonic anhydrase 1, chloroplastic (AtbCA1; AtbetaCA1)	BCA1_ARATH	3	4	36.7	7.43
GLYMA_19G147400	Delta(12)-fatty-acid desaturase	FAD6E_ARATH	2	5	43.8	8.46

PSM, peptide spectrum matches; pI, isoelectric point.

**Table 3 ijms-23-13270-t003:** Up-regulated membrane proteins under salt stress in soybean roots.

Ensembl Protein ID	Primary Protein Name	Arabidopsis Identifier	No. of Peptides	No. of PSMs	MW [kDa]	Calc. pI
GLYMA_01G181700	Probable serine/threonine-protein kinase PBL19	PBL19_ARATH	1	1	41.9	8.73
GLYMA_02G287000	Peroxisomal and mitochondrial division factor 2	PMD2_ARATH	16	49	34.9	5.1
GLYMA_04G174800	ATPase 11, plasma membrane-type	PMA11_ARATH	19	62	105.2	6.73
GLYMA_04G240400	Vacuolar protein sorting-associated protein 45 homolog (AtVPS45)	VPS45_ARATH	1	2	64.9	6.77
GLYMA_05G078000	Wall-associated receptor kinase-like 6	WAKLF_ARATH	1	1	25.2	9.41
GLYMA_05G215100	COP1-interactive protein 1	CIP1_ARATH	8	13	139.4	4.94
GLYMA_06G161400	Bidirectional sugar transporter SWEET15 (AtSWEET15)	SWT15_ARATH	4	8	68.7	5.67
GLYMA_08G020300	V-type proton ATPase subunit E1 (V-ATPase subunit E1)	VATE1_ARATH	21	141	26.8	6.25
GLYMA_08G021400	COP1-interactive protein 1	CIP1_ARATH	1	1	162.5	5.01
GLYMA_08G100800	Probable LRR receptor-like serine/threonine-protein kinase At1g67720	Y1677_ARATH	1	2	104.3	6.68
GLYMA_08G348200	Alpha-mannosidase I MNS4	MNS4_ARATH	2	22	6.8	9.2
GLYMA_10G024000	Phospholipid:diacylglycerol acyltransferase 1 (AtPDAT)	PDAT1_ARATH	1	2	21.6	10.18
GLYMA_10G107700	Calcium-transporting ATPase 4, plasma membrane-type	ACA4_ARATH	8	30	113.9	6.37
GLYMA_10G132300	Ammonium transporter 1 member 1 (AtAMT1;1)	AMT11_ARATH	1	5	53.3	7.3
GLYMA_10G218800	Somatic embryogenesis receptor kinase 1 (AtSERK1)	SERK1_ARATH	2	8	68.9	5.73
GLYMA_10G220100	Ras-related protein RABG3a (AtRABG3a)	RAG3A_ARATH	3	5	23.1	5.68
GLYMA_11G146900	Receptor-like protein 4 (AtRLP4)	RLP4_ARATH	1	2	16.4	9.6
GLYMA_11G250100	Mitochondrial import receptor subunit TOM20-2	TO202_ARATH	2	9	22.5	5.21
GLYMA_12G028300	Delta(24)-sterol reductase	DIM_ARATH	11	81	66	8.22
GLYMA_13G101400	Caffeoylshikimate esterase	CSE_ARATH	1	1	36.7	7.06
GLYMA_14G126300	Cytochrome b-c1 complex subunit 7-2	QCR72_ARATH	16	170	14.6	9.61
GLYMA_15G072100	Clathrin light chain 1	CLC1_ARATH	31	139	35.3	5.19
GLYMA_15G222300	Calcium-dependent protein kinase 2	CDPK2_ARATH	1	1	58.8	5.77
GLYMA_16G154200	Putative syntaxin-131 (AtSYP131)	SY131_ARATH	2	4	34.8	7.94
GLYMA_18G141900	High affinity nitrate transporter 2.5 (AtNRT2:5)	NRT25_ARATH	1	10	55.5	9.2
GLYMA_20G217100	Early nodulin-like protein 1	ENL1_ARATH	1	3	21.5	7.46

PSM, peptide spectrum matches; pI, isoelectric point.

**Table 4 ijms-23-13270-t004:** Down-regulated membrane proteins under salt stress in soybean roots.

Ensembl Protein ID	Primary Protein Name	Arabidopsis Identifier	No. of Peptides	No. of PSMs	MW [kDa]	Calc. pI
GLYMA_01G007700	Mitochondrial import inner membrane translocase subunit TIM17-2	TI172_ARATH	4	14	20.9	5.12
GLYMA_02G086600	Binding partner of ACD11 1	BPA1_ARATH	2	2	33.7	5.45
GLYMA_02G307700	Fasciclin-like arabinogalactan protein 7	FLA7_ARATH	2	6	27.1	5.29
GLYMA_02G309400	Transmembrane 9 superfamily member 12	TMN12_ARATH	2	4	75.1	6.86
GLYMA_03G190500	Bifunctional enolase 2/transcriptional activator	ENO2_ARATH	10	45	47.6	5.69
GLYMA_04G095900	Tetratricopeptide repeat domain-containing protein PYG7, chloroplastic	PYG7_ARATH	7	20	24.1	8.7
GLYMA_04G181000	Probable cyclic nucleotide-gated ion channel 16	CNG16_ARATH	1	1	79.4	9.16
GLYMA_04G220400	Probable inactive receptor kinase At5g10020	Y5020_ARATH	1	1	113.6	7.11
GLYMA_04G222800	Probable LRR receptor-like serine/threonine-protein kinase IRK	IRK_ARATH	1	1	104.7	6.61
GLYMA_05G016100	CHAPERONE-LIKE PROTEIN OF POR1, chloroplastic (AtCPP1)	CPP1_ARATH	1	1	28.3	9.88
GLYMA_05G199200	Calnexin homolog 1	CALX1_ARATH	4	14	61.6	4.91
GLYMA_06G019400	ATP-dependent zinc metalloprotease FTSH 5, chloroplastic (AtFTSH5)	FTSH5_ARATH	6	13	74.3	6.24
GLYMA_06G066400	ER membrane protein complex subunit 7 homolog	Y4213_ARATH	11	47	23.3	8.76
GLYMA_07G014400	Probable mitochondrial-processing peptidase subunit alpha-1, mitochondrial	MPPA1_ARATH	17	76	54.4	6.27
GLYMA_07G059400	SUN domain-containing protein 1 (AtSUN1)	SUN1_ARATH	1	7	8.5	6.54
GLYMA_08G348000	Galacturonosyltransferase 8	GAUT8_ARATH	1	1	64.1	9.16
GLYMA_09G035300	ABC transporter G family member 19 (ABC transporter ABCG.19; AtABCG19)	AB19G_ARATH	9	47	12.2	5.07
GLYMA_09G114500	Exocyst complex component SEC3A (AtSec3a)	SEC3A_ARATH	1	2	101	5.87
GLYMA_09G243700	Subtilisin-like protease SBT5.4	SBT54_ARATH	5	10	84.3	9.14
GLYMA_10G208300	Ras-related protein RABF1 (AtRABF1)	RABF1_ARATH	1	4	21.7	7.14
GLYMA_10G211000	Aquaporin PIP2-2	PIP22_ARATH	1	2	31.7	7.85
GLYMA_11G122300	Mitochondrial import receptor subunit TOM6 homolog	TOM6_ARATH	2	4	6.2	9.61
GLYMA_11G142300	Ras-related protein RABE1c (AtRABE1c)	RAE1C_ARATH	10	35	23.6	7.83
GLYMA_12G134800	Early nodulin-like protein 1	ENL1_ARATH	2	7	22.8	6.92
GLYMA_12G205400	ADP, ATP carrier protein 3, mitochondrial	ADT3_ARATH	19	123	39.4	9.79
GLYMA_13G058400	Membrane-anchored ubiquitin-fold protein 2 (AtMUB2)	MUB2_ARATH	3	7	12.6	6.09
GLYMA_14G076000	Outer envelope pore protein 24B, chloroplastic	OP24B_ARATH	14	83	23.7	9.26
GLYMA_14G201500	Protein TIC110, chloroplastic	TI110_ARATH	10	35	109.8	6.19
GLYMA_15G205900	Temperature-induced lipocalin-1 (AtTIL1)	TIL_ARATH	3	9	21.2	8.07
GLYMA_16G166600	2-succinyl-6-hydroxy-2,4- cyclohexadiene-1-carboxylate synthase	PHYLO_ARATH	3	9	12.6	6.86
GLYMA_17G021200	Probable prolyl 4-hydroxylase 4 (AtP4H4)	P4H4_ARATH	11	44	33	7.24
GLYMA_20G006500	Proton pump-interactor 1	PPI1_ARATH	3	4	161.1	4.51

PSM, peptide spectrum matches; pI, isoelectric point.

**Table 5 ijms-23-13270-t005:** Salt-stress-inducible membrane proteins in soybean leaves.

Ensembl Protein ID	Primary Protein Name	Arabidopsis Identifier	No. of Peptides	No. of PSMs	MW [kDa]	Calc. pI
GLYMA_05G019400	ABC transporter B family member 27 (ABC transporter ABCB.27; AtABCB27)	AB27B_ARATH	1	1	68.5	8.46
GLYMA_01G194500	ADP-ribosylation factor-like protein 8a (AtARL8a)	ARL8A_ARATH	4	5	20.5	8.15
GLYMA_16G076500	AP-4 complex subunit epsilon	AP4E_ARATH	1	1	107.4	5.92
GLYMA_12G172500	Aquaporin PIP2-2	PIP22_ARATH	1	2	30.8	8.13
GLYMA_09G056300	ATPase 2, plasma membrane-type	PMA2_ARATH	24	44	112.4	6.42
GLYMA_04G203800	Bidirectional sugar transporter SWEET15 (AtSWEET15)	SWT15_ARATH	2	2	67.1	6.11
GLYMA_06G324300	Cellulose synthase-like protein G1 (AtCslG1)	CSLG1_ARATH	4	8	79.5	8.57
GLYMA_12G095700	Chaperone protein dnaJ 2 (AtDjA2)	DNAJ2_ARATH	5	7	46.3	6.28
GLYMA_17G030700	COBRA-like protein 7	COBL7_ARATH	1	1	70.4	6.89
GLYMA_07G202300	Cytochrome P450 705A20	C75AK_ARATH	2	2	58.9	9.01
GLYMA_07G189900	External alternative NAD(P)H-ubiquinone oxidoreductase B2, mitochondrial	NDB2_ARATH	2	4	65.2	7.65
GLYMA_09G267700	Fasciclin-like arabinogalactan protein 10	FLA10_ARATH	14	41	42.5	5.97
GLYMA_12G069300	Fasciclin-like arabinogalactan protein 11	FLA11_ARATH	1	2	27.8	8.88
GLYMA_02G308700	Fasciclin-like arabinogalactan protein 2	FLA2_ARATH	9	23	43.5	6.87
GLYMA_02G307700	Fasciclin-like arabinogalactan protein 7	FLA7_ARATH	2	2	27.1	5.29
GLYMA_05G032500	GDP-mannose transporter GONST2	GONS2_ARATH	1	1	6.1	7.24
GLYMA_11G118500	Guanine nucleotide-binding protein subunit beta	GBB_ARATH	1	4	41	7.27
GLYMA_06G188900	HIPL1 protein	HIPL1_ARATH	1	2	75	5.17
GLYMA_05G029800	Hypersensitive-induced response protein 1 (AtHIR1)	HIR1_ARATH	11	18	31.3	7.01
GLYMA_15G248900	Interactor of constitutive active ROPs 1	ICR1_ARATH	1	1	42	5.76
GLYMA_02G079400	Light-harvesting complex-like protein OHP2, chloroplastic	OHP2_ARATH	5	12	19.8	8.87
GLYMA_14G099800	Mitochondrial carrier protein CoAc1	COAC1_ARATH	1	2	29.6	10.04
GLYMA_08G115600	Mitochondrial dicarboxylate/tricarboxylate transporter DTC	DTC_ARATH	8	20	32	9.35
GLYMA_13G096000	Mitochondrial outer membrane protein porin 4	VDAC4_ARATH	8	22	29.7	8.57
GLYMA_19G101100	Mitochondrial phosphate carrier protein 3, mitochondrial	MPCP3_ARATH	6	14	39.8	9.26
GLYMA_09G114000	Outer envelope pore protein 24B, chloroplastic	OP24B_ARATH	17	69	23.7	9.26
GLYMA_18G297700	Piezo-type mechanosensitive ion channel homolog	PIEZO_ARATH	2	2	21.1	9.85
GLYMA_06G020400	Plastocyanin minor isoform, chloroplastic	PLAS1_ARATH	1	2	16.8	5.2
GLYMA_20G141900	Probable dolichyl-diphosphooligosaccharide-subunit 3B	OST3B_ARATH	2	2	37.8	9.25
GLYMA_15G064700	Probable NAD(P)H dehydrogenase (quinone) FQR1-like 1	FQRL1_ARATH	17	37	21.7	6.95
GLYMA_13G212700	Probable prolyl 4-hydroxylase 4 (AtP4H4)	P4H4_ARATH	9	21	33.3	5.81
GLYMA_08G288900	SUPPRESSOR OF QUENCHING 1, chloroplastic	SOQ1_ARATH	2	3	44.1	8.41
GLYMA_13G307600	Putative syntaxin-131 (AtSYP131)	SY131_ARATH	6	8	34.5	7.49
GLYMA_07G028500	Pyrophosphate-energized vacuolar membrane proton pump 1	AVP1_ARATH	2	5	80.7	5.58
GLYMA_11G118800	Ras-related protein RABG3f (AtRABG3f)	RAG3F_ARATH	2	3	23	5.19
GLYMA_11G146900	Receptor-like protein 4 (AtRLP4)	RLP4_ARATH	1	1	16.4	9.6
GLYMA_07G072100	Serine/threonine-protein kinase BSK3	BSK3_ARATH	1	1	55.2	6.1
GLYMA_02G275500	SNF1-related protein kinase regulatory subunit beta-1 (AKIN subunit beta-1; AKINB1; AKINbeta1)	KINB1_ARATH	1	1	30.8	6.32
GLYMA_04G246100	Sucrose transport protein SUC9	SUC9_ARATH	1	1	14.8	9.25
GLYMA_17G152900	Tobamovirus multiplication protein 2A (AtTOM2A)	TOM2A_ARATH	2	6	31.5	5.36
GLYMA_10G243500	Transmembrane emp24 domain-containing protein p24beta3	P24B3_ARATH	2	2	23.9	7.09
GLYMA_19G093900	Triacylglycerol lipase SDP1	SDP1_ARATH	1	4	8.5	9.47
GLYMA_03G214700	UDP-glucuronic acid decarboxylase 2	UXS2_ARATH	10	29	48.1	7.97
GLYMA_14G035500	Vesicle-associated membrane protein 713 (AtVAMP713)	VA713_ARATH	3	4	25	9.42
GLYMA_20G014300	Auxin efflux carrier component 3 (AtPIN3)	PIN3_ARATH	1	1	72.2	7.64
GLYMA_16G038800	Binding partner of ACD11 1	BPA1_ARATH	2	4	29.9	9.11
GLYMA_05G204300	Gamma carbonic anhydrase-like 1, mitochondrial (AtCAL1; GAMMA CAL1)	GCAL1_ARATH	9	42	27.7	8.18
GLYMA_09G009300	Mitochondrial pyruvate carrier 4	MPC4_ARATH	1	4	10.5	10.36
GLYMA_09G271600	NADH dehydrogenase [ubiquinone] 1 alpha subcomplex subunit 1	NDUA1_ARATH	4	27	7.3	8.88
GLYMA_13G212700	Probable prolyl 4-hydroxylase 4 (AtP4H4)	P4H4_ARATH	6	11	33.3	5.81
GLYMA_05G150300	Protein TIC 22-like, chloroplastic	TI22L_ARATH	2	2	28.2	8.4
GLYMA_01G036200	RAN GTPase-activating protein 1 (AtRanGAP1; RanGAP1)	RAGP1_ARATH	1	1	65.1	8.07
GLYMA_16G090700	RPM1-interacting protein 4	RIN4_ARATH	1	1	27.1	8.29
GLYMA_19G093900	Triacylglycerol lipase SDP1	SDP1_ARATH	1	3	8.5	9.47
GLYMA_07G008600	UDP-glucuronate 4-epimerase 4	GAE4_ARATH	1	1	14.6	11.84
GLYMA_14G043500	Very-long-chain enoyl-CoA reductase	TECR_ARATH	3	9	35.9	9.63
GLYMA_05G214200	V-type proton ATPase subunit E3 (V-ATPase subunit E3)	VATE3_ARATH	10	22	27.6	6.8

PSM, peptide spectrum matches; pI, isoelectric point.

**Table 6 ijms-23-13270-t006:** Salt-stress-inducible membrane proteins in soybean roots.

Ensembl Protein ID	Primary Protein Name	Arabidopsis Identifier	No. of Peptides	No. of PSMs	MW [kDa]	Calc. pI
GLYMA_13G220000	ABC transporter C family member 2 (ABC transporter ABCC.2; AtABCC2)	AB2C_ARATH	2	2	181.7	6.61
GLYMA_11G098800	ABC transporter G family member 8 (ABC transporter ABCG.8; AtABCG8)	AB8G_ARATH	1	4	21.2	9.44
GLYMA_12G094000	Alternative NAD(P)H-ubiquinone oxidoreductase C1, chloroplastic/mitochondrial	NDC1_ARATH	4	29	39.5	8.19
GLYMA_10G167800	Ammonium transporter 1 member 2 (AtAMT1;2)	AMT12_ARATH	1	2	53.6	7.44
GLYMA_11G119800	Arabinosyltransferase RRA3	RRA3_ARATH	1	1	49.4	8.78
GLYMA_04G162800	ATP synthase subunit gamma, mitochondrial	ATPG3_ARATH	9	42	35.2	9.29
GLYMA_06G019400	ATP-dependent zinc metalloprotease FTSH 5, chloroplastic (AtFTSH5)	FTSH5_ARATH	3	4	74.3	6.24
GLYMA_16G204600	Bifunctional enolase 2/transcriptional activator	ENO2_ARATH	4	17	48	6.47
GLYMA_16G038800	Binding partner of ACD11 1	BPA1_ARATH	2	4	29.9	9.11
GLYMA_08G180800	AtBAK1; BRI1-associated receptor kinase 1	BAK1_ARATH	1	2	67.9	5.59
GLYMA_12G095700	Chaperone protein dnaJ 2 (AtDjA2)	DNAJ2_ARATH	5	15	46.3	6.28
GLYMA_12G040500	Coatomer subunit delta	COPD_ARATH	1	2	58.4	5.59
GLYMA_11G118600	Coatomer subunit gamma	COPG_ARATH	4	7	98.7	5.15
GLYMA_11G063400	Cystinosin homolog	CTNS_ARATH	1	2	42.9	8.12
GLYMA_05G110100	Cytochrome c oxidase subunit 6a, mitochondrial (AtCOX6a)	COX6A_ARATH	5	41	11.5	9.09
GLYMA_17G157000	Cytochrome c oxidase subunit 6a, mitochondrial (AtCOX6a)	COX6A_ARATH	5	41	11.6	9.35
GLYMA_06G176200	Cytochrome P450 71A22	C71AM_ARATH	1	1	58.5	7.36
GLYMA_10G286200	Dihydroorotate dehydrogenase (quinone), mitochondrial (DHOdehase)	PYRD_ARATH	1	2	48.3	9.31
GLYMA_13G171700	Dolichol-phosphate mannosyltransferase subunit 1	DPM1_ARATH	1	1	29.3	9.54
GLYMA_09G129900	DUF21 domain-containing protein At5g52790	Y5279_ARATH	1	1	39.1	8.59
GLYMA_07G170200	E3 ubiquitin protein ligase RIE1	RIE1_ARATH	1	1	39.6	6.34
GLYMA_16G071500	EIN2-CEND	EIN2_ARATH	1	1	486.8	4.31
GLYMA_02G276600	Elongation factor 1-beta 1 (EF-1-beta 1)	EF1B1_ARATH	11	43	24.3	4.68
GLYMA_20G194100	Exocyst complex component EXO70A1 (AtExo70a1)	E70A1_ARATH	1	3	72.9	7.2
GLYMA_17G260400	Expansin-A11 (AtEXPA11)	EXP11_ARATH	1	2	27.5	8.84
GLYMA_08G239300	FT-interacting protein 1	FTIP1_ARATH	1	2	117.1	8.57
GLYMA_04G013100	Guanine nucleotide-binding protein subunit beta	GBB_ARATH	1	1	41	7.44
GLYMA_17G124900	High-affinity nitrate transporter 3.1	NRT31_ARATH	1	1	23.2	9.09
GLYMA_05G133700	LysM domain-containing GPI-anchored protein 2	LYM2_ARATH	1	3	40.2	7.24
GLYMA_10G057000	Membrane magnesium transporter	MMGT_ARATH	1	1	11.9	7.03
GLYMA_10G049900	Mitochondrial import inner membrane translocase subunit TIM14-1	TI141_ARATH	1	2	12.2	10.13
GLYMA_07G193700	Mitochondrial inner membrane protein OXA1	OXA1_ARATH	1	3	47.1	9.35
GLYMA_01G157300	Monosaccharide-sensing protein 3	MSSP3_ARATH	2	3	79.4	6.18
GLYMA_03G114600	Oxygen-evolving enhancer protein 3-2, chloroplastic (OEE3)	PSBQ2_ARATH	2	4	24.8	9.85
GLYMA_12G001900	Patellin-3	PATL3_ARATH	2	6	48.9	5.44
GLYMA_14G199000	Sorting nexin 2B	SNX2B_ARATH	3	5	62.3	5.5
GLYMA_08G180400	Sulfate transporter 1.3	SUT13_ARATH	1	3	71.9	9.32
GLYMA_10G195800	Sulfoquinovosyl transferase SQD2	SQD2_ARATH	1	1	55.4	6.83
GLYMA_13G185300	SUN domain-containing protein 1 (AtSUN1)	SUN1_ARATH	1	1	50.5	8.19
GLYMA_12G032400	Synaptotagmin-2	SYT2_ARATH	3	11	61.6	6.65
GLYMA_18G215900	Thioredoxin H9 (AtTrxh9)	TRXH9_ARATH	1	2	15.7	5.17
GLYMA_07G010300	Transmembrane 9 superfamily member 7	TMN7_ARATH	2	4	73.7	8.32
GLYMA_08G193400	Transmembrane 9 superfamily member 7	TMN7_ARATH	2	4	73.6	8.22
GLYMA_16G054600	Triacylglycerol lipase SDP1	SDP1_ARATH	2	4	8.7	9.7
GLYMA_03G214700	UDP-glucuronic acid decarboxylase 2	UXS2_ARATH	10	37	48.1	7.97
GLYMA_07G119500	Vacuolar protein-sorting-associated protein 11 homolog (AtVPS11)	VPS11_ARATH	1	2	109.4	6
GLYMA_15G044700	V-type proton ATPase subunit a1 (V-ATPase subunit a1)	VHAA1_ARATH	2	7	94.2	6.4
GLYMA_20G106500	V-type proton ATPase subunit B2 (V-ATPase subunit B2)	VATB2_ARATH	33	210	54.2	5.02
GLYMA_02G065500	V-type proton ATPase subunit C (V-ATPase subunit C)	VATC_ARATH	13	50	42.5	5.95

PSM, peptide spectrum matches; pI, isoelectric point.

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
