# Peer review of "Membrane Proteomic Profiling of Soybean Leaf and Root Tissues Uncovers Salt-Stress-Responsive Membrane Proteins"

_ijms, 2022, doi:10.3390/ijms232113270_

Round 1
Reviewer 1 Report
The manuscript provides a comprehensive description of the proteome profiles of soybean leaf and root tissues under salt-stress compared to control. The experimental design was straightforward yet informative, and the analyses conducted and reported are clear.
An improvement to the readability of the manuscript and to shorten it would be to move some or all of the tables to the Supplemental Materials.
A few typos: / suggestions for textual improvements
Page 3, line 100: insert "has" before "an inhibitory effect...
Page 11, line 236: events (plural)
Page 25, line 404: remove second comma
Page 30, line 496: remove "some"
Page 34, Author Contributions: correct ... "and the SC and SZ helped me" into ... "and SC and SZ provided assistance" or similar.
Author Response
Reviewer 1
Comment#1 An improvement to the readability of the manuscript and to shorten it would be to move some or all of the tables to the Supplemental Materials.
Answer#1 Thank you very much for your comments to improve this manuscript. The proteomic data in main tables represents the up-regulated, downregulated and proteins induced under salt stress which we have explained in results and discussion section. Since it is our core data, we prefer to keep it in the main text.
A few typos: / suggestions for textual improvements
Comment#2 Page 3, line 100: insert "has" before "an inhibitory effect...
Answer#2 revised.
Comment#3 Page 11, line 236: events (plural)
Answer#3 revised.
Comment#4 Page 25, line 404: remove second comma
Answer#4 revised.
Comment#5 Page 30, line 496: remove "some"
Answer#5 revised.
Comment#6 Page 34, Author Contributions: correct ... "and the SC and SZ helped me" into ... "and SC and SZ provided assistance" or similar.
Answer#6 revised.

Reviewer 2 Report
The authors used label-free quantitative proteomic techniques to detect these differentially expressed membrane proteins in leaves and roots in response to 50mM and 100 mM NaCl treatments in the salt-sensitive genotype, C08. The scientific finding contains partial scientific interest. However, the author should select at least one or two genes for further functional investigation.
Author Response
Reviewer 2
Comment#1
The authors used label-free quantitative proteomic techniques to detect these differentially expressed membrane proteins in leaves and roots in response to 50mM and 100 mM NaCl treatments in the salt-sensitive genotype, C08. The scientific finding contains partial scientific interest. However, the author should select at least one or two genes for further functional investigation.
Answer#1
Thank you very much for your comment. As you have reviewed the manuscript which is purely based on proteomic experiment of membrane proteins extracted, identified and quantified under salt stress in soybean from root and leave tissues. Our proteomic data was analyzed according to the previous protocols and found many of the important membrane proteins changing their abundance under salt stress. To further strengthen this proteomic data, we have verified the expression patterns of up-regulated and down-regulated membrane proteins using published RNA-seq data of the same genotype.
Regarding functional investigation of membrane proteins, we have made use of the existing literatures to support our data interpretation in the discussions. Some highlights are given below:
1) Na+/H+ exchanger proteins, synaptotagmin-1, V-ATPases membrane proteins and their functional investigations in previous studies were added (Lines# 499-509).
2) ATPases which were upregulated in roots upon salt stress treatment were supported by previous functional investigation (Lines#525-536).
3) Several mitochondrial and chloroplast membrane proteins were downregulated upon salt stress treatment were supported by previous functional investigations (Lines# 567-574).
In summary, this manuscript provides an overall view of membrane proteins expressed under salt stress. Based of previous literatures, some of the proteins are of known putative functions in stress responses, supporting the validity of our data. New membrane proteins identified in this study will be worthwhile for future functional studies, which may lead to a better understanding of salt stress responses. Detailed functional analysis of selected genes will require in depth studies that are beyond the scope of this manuscript.

Round 2
Reviewer 2 Report
I agree the authors' responses. Accept in present form.